

# Atmospheric QBO and ENSO indices with high vertical resolution from GNSS radio occultation temperature measurements

Hallgeir Wilhelmsen[1,2,3], Florian Ladstädter[1,3], Barbara Scherllin-Pirscher[4], and Andrea K. Steiner[1,2,3]

[1]Wegener Center for Climate and Global Change (WEGC), University of Graz, Graz, Austria
[2]FWF-DK Climate Change, University of Graz, Graz, Austria
[3]Institute for Geophysics, Astrophysics, and Meteorology/Institute of Physics, University of Graz, Graz, Austria
[4]Zentralanstalt für Meteorologie und Geodynamik (ZAMG), Vienna, Austria

*Correspondence to:* Hallgeir Wilhelmsen (hallgeir.wilhelmsen@uni-graz.at)

**Abstract.** We provide atmospheric temperature variability indices for the tropical troposphere and stratosphere based on Global Navigation Satellite System (GNSS) Radio Occultation (RO) temperature measurements. By exploiting the high vertical resolution and the uniform distribution of the GNSS RO temperature soundings we introduce two approaches, both based on an empirical orthogonal function (EOF) analysis. The first method utilizes the whole vertical and horizontal RO temperature field

from 30° S to 30° N and from 2 km to 35 km altitude. The resulting indices, the leading principle components, resemble the well-known patterns of the Quasi-Biennial Oscillation (QBO) and the El Niño-Southern Oscillation (ENSO) in the tropics. They provide some information on the vertical structure, however, they are not vertically resolved. The second method applies the EOF analysis on each altitude level separately and the resulting indices contain information on the horizontal variability at each densely available altitude level. They capture more variability than the indices from the first method and present a mixture

of all variability modes contributing at the respective altitude level, including the QBO and ENSO. Compared to commonly used variability indices from QBO winds or ENSO sea surface temperature, these new indices cover the vertical details of the atmospheric variability. Using them as proxies for temperature variability is also of advantage because there is no further need to account for response time lags. Atmospheric variability indices as novel products from RO are expected to be of high benefit for studies on atmospheric dynamics and variability, for climate trend analysis, as well as for climate model evaluation.

# 1 Introduction

Two modes of interannual variability dominate the natural temperature variability in the tropical region: The Quasi-Biennial Oscillation (QBO) and El Niño-Southern Oscillation (ENSO).

In the stratosphere, equatorial zonal wind regimes of easterlies (winds blowing from east) and westerlies propagate downwards at about 1 km/month. As soon as the westerlies fade out, the easterlies take over, and vice versa. The winds at 10 hPa

are out of phase with the winds at 70 hPa. The period of the regimes varies considerably, with an average period of a little more than two years (approximately 28 months), which has given the phenomenon its name, the QBO.

There are several characteristics describing the QBO. The two wind regimes do not change much along the longitudinal axis (Naujokat, 1986), but exhibit a distinct latitudinal structure. Strongest QBO-related winds are latitudinally symmetric, centered





over the equator (Dunkerton and Delisi, 1985), and decrease considerably off the equator with a meridional half width of less than 15°. The winds are strongest in the mid to lower tropical stratosphere, although the QBO is detectable from the tropopause up to 50 km (Wallace et al., 1993). For a review on QBO features, see Baldwin et al. (2001).

Recently there was a disruption in the stable QBO cycle, where the westerlies took a shortcut upwards, and prevented the
easterlies to propagate downwards to the troposphere, as it usually does (Newman et al., 2016; Osprey et al., 2016) which also manifested in a response of anomalous behavior of trace gases (Tweedy et al., 2017). The causes of this unprecedented disruption in the QBO observational record are still under investigation. Possible explanations include dynamical forcing from waves (Osprey et al., 2016; Coy et al., 2017) or coupling with the warm ENSO event 2015/2016 (Newman et al., 2016; Dunkerton, 2016).

In the troposphere, the dominant interannual variability mode is the ENSO. Its irregular variability origins from a Pacific ocean-atmosphere interaction and manifests itself as a warm phase (El Niño) and a cold phase (La Niña). Its effects can be detected globally, from the surface to the lower stratosphere (Free and Seidel, 2009; Randel et al., 2009). Commonly the characteristics are described with an ENSO sea surface temperature (SST) index, which can be derived from anomalies in SST in the Niño 3.4 region (5° N to 5° S and 170° W to 120° W) of the tropical Pacific. El Niño or La Niña periods are defined
to occur if a certain mean SST anomaly threshold is exceeded over several months, e.g., if 5-month running means of SST anomalies in the Niño 3.4 region exceed $+0.4°$ C or $-0.4°$ C, respectively, for a minimum of 6 months as defined by Trenberth (1997).

During an El Niño event, there is also a warming in the tropical troposphere, with a maximum around 8 km and a cooling in the lowermost stratosphere above the tropopause (Reid, 1994; Randel et al., 2009). There is also an eddy signal, i.e. deviations
from the zonal mean (Fernández et al., 2004), with a maximum around 11 km (Scherllin-Pirscher et al., 2012).

Several indices have been introduced to describe the variability of QBO and ENSO in order to better characterize the events, to discriminate their strength, and to describe their evolution. Trenberth and Stepaniak (2001) suggested the need for a second index, in addition to the Niño 3.4 SST index, to describe the different characteristics of the ENSO-originated variability. Wolter and Timlin (2011) established an extended multivariate ENSO index that is more complete and flexible compared to
single variable ENSO indices.

QBO characteristics can be exploited in the commonly used QBO indices, often derived from wind speeds, serving as proxies to describe the QBO. Naujokat (1986) established the well-known set of time series from winds at different pressure levels above Singapore. Most frequently used QBO indices are wind anomalies at 30 hPa and 50 hPa pressure levels. Wallace et al. (1993) introduced the representation of the equatorial stratospheric QBO in derived parameters using the leading empirical
orthogonal functions of the vertical wind structure.

In this work, we describe a method to create atmospheric variability indices with high vertical resolution, using Global Navigation Satellite System (GNSS) Radio Occultation (RO) satellite data.

GNSS RO is a limb sounding technique, where the GNSS signals traverse the Earth's atmosphere, and are picked up by receivers on Low Earth Orbit (LEO) satellites as they rise or set behind the Earth's horizon relative to the GNSS satellite. On
their way to the LEO satellites, the signals are refracted by the atmosphere as they propagate through it. The resulting excess





phase path is measured at the LEO satellite. With the help of inversion methods and prior knowledge about the atmosphere, the refraction can be inverted into several atmospheric parameters, one of which is temperature (Melbourne et al., 1994; Kursinski et al., 1997).

The GNSS RO measurements provide vertical atmospheric temperature profiles with a high vertical resolution that are well
distributed globally. There have been several RO missions throughout the years, providing data continuously since May 2001, and it has been shown that there is no need for calibration between the missions (Schreiner et al., 2007; Foelsche et al., 2011). GNSS RO measurements are of best quality in the upper troposphere and lower stratosphere region, at about 8 km to 35 km in the tropics (Scherllin-Pirscher et al., 2011).

Randel et al. (2003) and Schmidt et al. (2004) showed clear evidence that GNSS RO temperature anomalies can be used for
detecting the QBO. This was achieved with only few years of data in the early phase of the GNSS RO period.

Scherllin-Pirscher et al. (2012) were able to demonstrate that the structure of the ENSO can be detected with GNSS RO temperature anomalies, using only four years of data. They confirmed the spatial structure of the ENSO during the El Niño events, and showed that the zonal atmospheric response lags SST anomalies by 3 months.

Several other studies have investigated the atmospheric QBO and ENSO signal using GNSS RO data in analyses of climate
trends (e.g., Lackner et al., 2011; Steiner et al., 2011) and climate variability (e.g., Randel and Wu, 2015). Teng et al. (2013) and Sun et al. (2014) investigated signatures and characteristics of the ENSO while Gao et al. (2017) used RO measurements to create an index that describes the strength of the atmospheric response from ENSO and QBO.

We extend on this previous work, and use the whole available GNSS RO time series from 2001 to 2016 to create atmospheric variability proxies. We describe the GNSS RO data set in Sect. 2 and explain the applied methods in Sect. 3. Results are
presented and discussed in Sect. 4. A summary and conclusions are given in Sect. 5.

## 2  Data

We use data from the Wegener Center (WEGC) OPSv5.6 RO multi-satellite record (Schwärz et al., 2016; Angerer et al., 2017, in review) to produce monthly mean gridded temperature fields with a horizontal resolution of $5° \times 5°$ in longitude and latitude and 100 m vertical resolution. The time period ranges from from May 2001 to October 2016.

The WEGC OPSv5.6 data record is a data set with global coverage, but in order to mainly obtain the QBO and ENSO signals, we restrict it to 2 km to 35 km altitude and 30° S to 30° N in latitude. Grid points with missing data are filled horizontally using bilinear interpolation, at each time step, or filled with the nearest neighbor data at the boundaries. Our input data set used in this study therefore has $N_\phi = 12$ grid points in latitude ($\phi$), $N_\theta = 72$ grid points in longitude ($\theta$), $N_z = 331$ grid points in altitude ($z$), and $N_t = 186$ grid points in time ($t$).

We create monthly mean temperature anomalies to deseasonalize the data. This is done by calculating the average temperature for each month for the reference time period January 2002 to December 2015. These monthly averages are then subtracted from the original temperature data at the corresponding months. To reduce the month-to-month variations, the monthly mean





anomalies are then smoothed with a 1-2-1 running mean filter over time. Finally, time series at each grid point are detrended to avoid any trend in indices of atmospheric variability.

Figure 1 shows zonal mean RO temperature anomalies from 20° S to 20° N for illustration purposes. We clearly see the downward propagating QBO pattern in the lower stratosphere, known from previous work, where negative temperature anomalies
correspond to westerlies and positive temperature anomalies correspond to easterlies. The highest variability is attributed to the transition from westerlies to easterlies (Scherllin-Pirscher et al., 2017). In the troposphere, we see the variability from the ENSO phenomenon. Several El Niño events are revealed during the GNSS RO time period: During the winter in 2002–2003, 2004–2005, 2006–2007, 2009–2010 and a major event in 2014–2016, that lasted longer than normal (Blunden and Arndt, 2016). The La Niña events 2007–2008, 2010–2011, 2011–2012 can also be seen.

## 3   Methods

We create the atmospheric variability indices using two different methods, in the following denoted M1 and M2. They are described in more detail in Sect. 3.1 and Sect. 3.2, respectively.

In both methods the main variability patterns in the input data set are obtained using an empirical orthogonal function (EOF) analysis. The EOF analysis decomposes the data set into a reduced set of space components and time components (denoted as indices). The first few components will describe most of the variability in descending order of importance (Jolliffe, 2002;
Hannachi et al., 2007).

In the following, we use the terminology from Hannachi et al. (2007), where "EOF" denotes the spatial component, while "PC" denotes the time component, to describe the output from the EOF analysis. When needed, we use the whole word "principle component" to describe the collection of EOFs, PCs, and eigenvalues.

### 3.1   EOF analysis on the whole temperature field (M1)

In the first method, denoted M1, a space-time matrix is constructed from the monthly mean temperature anomalies described in Sect. 2. The space dimensions are all stringed along a single axis, $(\phi, \theta, z)$, to reduce the number of dimensions. The resulting matrix is therefore a two dimensional matrix, in space $(s)$ and time $(t)$ only, $X(s_{(\phi,\theta,z)}, t)$, represented by Eq. (1).

$$
\boldsymbol{X} =
\begin{pmatrix}
x_{(s_1,t_1)} & x_{(s_1,t_2)} & \cdots & x_{(s_1,t_q)} & \cdots & x_{(s_1,t_Q)} \\
x_{(s_2,t_1)} & x_{(s_2,t_2)} & \cdots & x_{(s_2,t_q)} & \cdots & x_{(s_2,t_Q)} \\
\vdots & \vdots & \ddots & \vdots & \ddots & \vdots \\
x_{(s_p,t_1)} & x_{(s_p,t_2)} & \cdots & x_{(s_p,t_q)} & \cdots & x_{(s_p,t_Q)} \\
\vdots & \vdots & \ddots & \vdots & \ddots & \vdots \\
x_{(s_P,t_1)} & x_{(s_P,t_2)} & \cdots & x_{(s_P,t_q)} & \cdots & x_{(s_P,t_Q)}
\end{pmatrix}
\tag{1}
$$





Each spatial location is denoted with index $s_p$, where $p = 1 \ldots P$. Each point in time is denoted $t_q$, where $q = 1 \ldots Q$. For M1, using the input data set described in Sect. 2, the spatial direction has the length, $P = N_\phi \cdot N_\theta \cdot N_z = 285984$. The time dimension has the length $Q = 186$.

Each row, $\boldsymbol{x}_{s_p}(t)$, which represents a time series at a specific location, is *centralized* by subtracting the arithmetic mean of the time series at each grid point. The EOF analysis is then based on the decomposition of the covariance matrix (Jolliffe, 2002).

The output of the EOF analysis is a set of EOFs (EOF$_{\text{out}}$), PCs (PC$_{\text{out}}$), and their corresponding eigenvalues ($\lambda$). The eigenvalues are used to scale the corresponding output EOFs and PCs, according to both Eq. (2) and Eq. (3):

$$\text{EOF}_i = \text{EOF}_{\text{out},i} \sqrt{\boldsymbol{\lambda}_i} \tag{2}$$

$$\text{PC}_i = \frac{\text{PC}_{\text{out},i}}{\sqrt{\boldsymbol{\lambda}_i}} \tag{3}$$

The first few PCs from Eq. (3) can now be used as proxies for the temporal variability, which we call indices in the following.

### 3.2 EOF analysis at each altitude level (M2)

To take advantage of the high vertical resolution from RO we also calculate atmospheric variability indices at all altitude levels. In this second method, denoted M2, we do the EOF analysis for each altitude level separately, instead of using the whole field. No altitude dependent variability is therefore included in the analysis.

To keep the altitude dimension separated from the other dimensions, a space-time matrix is constructed for each altitude level. Therefore, only the latitude and longitude dimensions are stringed along one axis, leaving us with the space $(\phi, \theta)$, and time $(t)$ dimensions, leading to matrix $X_z(s_{(\phi,\theta)}, t)$, which is represented by Eq. (1) at each altitude level, $z$.

The spatial direction has the length $P = N_\phi \cdot N_\theta = 864$ and the time dimension has the length $Q = 186$. The subsequent steps in the EOF analysis are the same as for M1.

## 4 Results

The values within each independent EOF show *where* the principle component contributes to the variability, and how much each point is influenced by its corresponding PC. In the same way, each independent PC shows *when* the corresponding EOF changes.

In this section, we compare the EOFs and PCs constructed from M1 and M2 with characteristics of known atmospheric variability patterns.

### 4.1 M1 results

The first four resulting EOFs from M1 are presented in Fig. 2. Each EOF has been reshaped to the same space dimensions as the input data set, $(\phi, \theta, z)$. Each column shows an EOF at selected altitude levels.



The spatial structure of the variability from the first and the second EOF (EOF1 and EOF2, respectively) show characteristics of the QBO. From the stratosphere to the tropopause at around 17 km, the two EOFs do not show distinct longitudinal variability. The patterns are strongest over the equator with a symmetrical latitudinal dependency, and the tropical band varies antipodal to the extratropical latitude band. These features can also be observed with the QBO winds (Naujokat, 1986; Wallace et al., 1993; Baldwin et al., 2001).

EOF1 and EOF2 both exhibit only a weak signal at and below the tropopause, where the pattern also looks different than above.

This pattern around the tropopause is also visible in the third and the fourth EOFs (EOF3 and EOF4 respectively). The longitudinal pattern disappears at around 14 km where zonal variability dominates. However, it reappears with opposite sign further below at 11 km. This vertical behavior resembles quite well with the results of Scherllin-Pirscher et al. (2012) who found a strong eddy ENSO signal with a node at approximately 14 km. This eddy ENSO signal is superimposed with a zonal-mean ENSO signal, which has its node at 17 km.

EOF3 and EOF4 also contribute to the variability above the tropopause. Although the pattern is weak, it is interesting that the signals resemble the patterns of EOF1 and EOF2 above the tropopause.

Figure 3 shows the corresponding PC time series (indices) to the EOFs. The two first PCs, PC1 and PC2, have a regular, sine wave like pattern, where one follows the other, with a period of about 2 years. PC3 and PC4 change more rapidly and less regularly.

The output patterns from M1 are not sensitive to the vertical resolution of the input field. We pruned the data to only include every 1 km before performing M1. The EOF pattern looked very much the same as described above, except the PCs showed a more coarse pattern, and the explained variances were a little smaller (not shown).

## 4.2 M2 results

Figure 4 shows the set of the two first EOFs from M2 at the same selected altitude levels as in Fig. 2. Remember, that while the EOFs of M1 are functions of latitude, longitude, and altitude, there are separate EOFs for each altitude level resulting from M2. All of these separate EOFs are functions of latitude and longitude only. The recomposed time series of the two first PCs are presented in Fig. 5. The first set of PCs reveals a separation into a part above the tropopause and into a part below the tropopause. Above the tropopause, the first PCs show the typical downward propagating QBO pattern. Below the tropopause, the PCs are associated with ENSO events.

The separation at the tropopause can also be seen in Fig. 4. Above the tropopause, the EOF1s from M2 resemble the patterns of either EOF1 or EOF2 of M1 shown in Fig. 2. Above the tropopause the EOF2s also show the same horizontal QBO pattern, though weaker. Below the tropopause, the horizontal patterns of EOF1 and EOF2 resemble EOF3 and EOF4 of M1, which we identified as ENSO-related variability.



### 4.3 Explained variance

Figure 6 shows *how much* each principle component contributes to the total variability. For M1, the first four principle components sum up to 57 % of the total variability, (top panel of Fig. 6). Remaining natural variability, associated with, e.g., volcanic eruptions or sudden stratospheric warming (SSW) events (Randel and Wu, 2015), as well as some remaining sampling issues

from GPS RO, account for 43 % of the variance.

For M2 (bottom panel of Fig. 6) the explained variance ratios are shown for the first three principal components at each altitude. Most of the variability is explained by the first principle component, except near the tropopause region, where the first and the second principle components almost touch. In the stratosphere and the troposphere the EOF analysis captures the dominant variability of the QBO and the ENSO, respectively. The tropopause region is less uniform. It is a transition layer

between the troposphere and the stratosphere, and more complex in its nature (Gettelman and Forster, 2002), which could be an explanation for the lower explained variance ratio.

### 4.4 Correlations to conventional indices

In order to show that our deduced indices capture the QBO and ENSO we compute the correlations between the derived PCs and conventional SST and QBO indices.

Figure 7 shows the correlations between the PCs from M1 with the conventional QBO wind indices at 30 hPa (QBO30), 50 hPa (QBO50), and the Niño 3.4 SST index. The correlations to the solar F10.7 cm flux index are also shown. We do not smooth nor detrend these indices.

We find that both QBO30 and QBO50 correlate well with PC1 and PC2 (top), and less well with PC3 and PC4 (bottom). The time lags are a result of the PCs representing the variation at altitudes specified by the EOF patterns. They therefore introduce

a time lag when correlating to wind fields at only two pressure levels.

PC3 and PC4 correlate best with the Niño 3.4 SST index. When PC3 is superimposed with PC4, the total correlation to the Niño 3.4 SST index is improved with a maximum correlation of 0.81 at a time lag of 3 months (not shown).

Figure 8 shows how the PC1s and PC2s derived from M2 correlate with the QBO30 index, QBO50 index, Niño 3.4 SST index, and solar F10.7 cm flux index at each altitude level.

The recomposed set of PC1s correlates well with the QBO30 and QBO50 above the tropopause, and Niño 3.4 SST index below the tropopause, depending on altitude and time lag. There is also some correlation between the set of PC2s to the same indices, although weaker, and without a clear pattern. Both the PC1s and the PC2s have only weak correlation to the solar F10.7 cm flux index.

### 4.5 Phase space diagram

In order to show the relationship between the PCs we present phase space diagrams in Fig. 9, following Wallace et al. (1993, Fig. 5 therein). Figure 9 (left panel) shows the relationship between the two first PCs from QBO winds as a trajectory in phase space (PC1 versus PC2). For comparison purposes M1 has been performed using QBO winds at seven pressure levels from





70 hPa to 10 hPa after Wallace et al. (1993). We use the same time period as available in the WEGC OPSv5.6 data set from May 2001 to October 2016. Before plotting we apply a 5 month running mean on the PCs. The resulting phase plot confirms the long history of circularity and nearly constant amplitude of the QBO. The QBO disruption that has been observed during 2016 can be clearly seen from the winds (Dunkerton, 2016), and it seems to have found its way back to normal by the end of

2016 (Tweedy et al., 2017).

Figure 9 (middle) shows the same as the left plot, but is constructed from RO temperature anomalies, using PC1 and PC2 from M1 (cf. Fig. 3 (upper two panels)). It has a similar structure and features as the phase plot from the winds. The QBO disruption is also revealed here, but it has not ended yet in temperature space. This further supports our findings that the main variability obtained by EOF1 and EOF2 from M1 is the QBO.

In Fig. 9 (right), a phase plot of two PC1s from M2, at two selected altitude levels in the QBO region, are shown. It does not show exactly the same circularity as seen in the two other plots, which could be a result from not covering the same altitude range as in the two other plots. Nevertheless the recent disruption of the QBO can also be seen here.

## 4.6 Reconstruction of temperature fields

The actual contribution from each principle component to the resulting temperature anomaly field can be seen when recon-

structing the principle components. We do this by multiplying the EOF with its corresponding PC. Any scaling by the eigenvalues, or sign flipping, is then canceled out.

Figure 10 (top left panel) shows the reconstructed field using a combination of the first and the second principle components from M1. We see that most of the contribution to the resulting temperature anomaly field is in the QBO region, which is also expected from the pattern of the EOFs. We clearly see the downward propagating pattern of the QBO, and only a weak signal

in the troposphere. It should be noted that the downward propagating pattern cannot be created by one principle component alone.

Figure 10 (middle left panel) shows the reconstruction using a combination of the third and the fourth principle component from M1. In the troposphere region we see a positive contribution to the temperature field during the El Niño events. The signal right above the tropopause might also be associated with El Niño (e.g., Scherllin-Pirscher et al., 2012) but further up,

the features seem to be related to the QBO.

Figure 10 (bottom left panel) shows a combination of all four principle components from M1, and the result well resembles the input temperature anomalies shown in Fig. 1.

For M2, temperature anomaly time series have been reconstructed separately at each altitude level from the resulting principle components. Figure 10 (top right panel) shows the reconstruction using the first principle component of each altitude level.

The result reveals that much of the features in Fig. 1 are already obtained.

Similarly, Fig. 10 (middle right panel) shows the reconstructions from the second principle component for each altitude layer from M2. It describes some features of the variability in the QBO region, that are not caught by the first principle component. The stratospheric variability pattern seen in the first principle component (Fig. 10, top right panel) seems to find its continuation in the second principle component (Fig. 10, middle right panel) just above the tropopause.




This suggests that the first principle components are attributable to different variabilities with altitude, and that the attribution can swap between the principle components. It is therefore important to include both indices to catch the variability, especially around the tropopause.

Figure 10 (bottom right panel) shows a combination of the first and second principle components, and as for M1, it also resembles the input temperature field (cf. Fig. 1) very well.

The resulting difference between the input field (Fig. 1) minus the respective reconstructed fields (Fig. 10, bottom) from M1 and M2 are presented in Fig. 11. This therefore describes the residue of the two methods. For M1 there is still some residue temperature variability left, especially in the tropopause region, but also in the other regions. For M2, however, there is only some minor residue left, especially in the tropopause region. M2 shows a much smaller residue than M1 which indicates that the altitude resolved indices better capture the atmospheric variability.

## 5 Summary and conclusions

Atmospheric variability in the tropical region is often described by indices of the two main modes of variability, the QBO and the ENSO. These indices, commonly derived from stratospheric winds and SST anomalies, do not cover the vertical details. Since they are not derived from atmospheric temperatures, we need to account for a potentially unknown time lag when using them as proxies for atmospheric temperature variability at a specific altitude level.

In this work we introduce new atmospheric variability proxies constructed directly from GNSS RO temperature measurements of high vertical resolution, using standard EOF analysis. We prepared the GNSS RO temperature field for the EOF analysis in two ways.

In the first method, the input field for the EOF analysis includes the whole vertical and horizontal information from 2 km to 35 km and from 30° S to 30° N. The resulting principle components show the well-known characteristics of QBO and ENSO, seen in previous work, and the first four PC time series describe the major part of the variability as a whole. However, they still contain an unknown time lag to the actual variability at different altitude levels, and do not show a strong dependency on the vertical resolution of the input field.

In the second method, we take advantage of the high vertical resolution of GNSS RO and perform an EOF analysis at each altitude level separately to obtain the main horizontal variability at each altitude level. These variability indices, which hold the high vertical resolution from RO, also show the well known characteristics of the QBO and ENSO, and as they are obtained directly where the variability occurs, they (by their nature) contain no time lag to the actual variability. However, the resulting PCs cannot be attributed to one or the other mode of variability only, but instead present a mixture of all variability modes found at the respective altitude level.

We find that the altitude resolved indices of the second method capture more of the atmospheric variability than the indices derived from the first method.



Testing the correlation with known classical sea surface temperature indices and wind indices confirmed that the indices derived from RO temperature represent atmospheric variability indices. Further confidence on the results is given as we find the characteristic relationship over time between PC time series from RO temperature consistent with those in winds.

We thus demonstrated that information on the most significant modes of natural climate variability in the tropical troposphere

and stratosphere can be derived from GNSS RO temperature observations. Taking advantage of these results, we derive novel products from RO with high added value. We provide vertically high resolved atmospheric variability indices which can deliver improved information on the natural variability patterns such as QBO and ENSO. Good representation and better knowledge of atmospheric and climate variability is of importance for studies of atmospheric physics and dynamics, the analyses of climate variability and trends, as well as for the evaluation of climate models.

*Data availability.* The WEGC OPSv5.6 RO dataset is available on request from the authors and will be made publicly available soon.

The QBO wind indices were downloaded from:

http://www.geo.fu-berlin.de/en/met/ag/strat/produkte/qbo.

The Niño 3.4 SST index were downloaded from:

http://www.cpc.ncep.noaa.gov/data/indices/sstoi.indices.

The solar F10.7 cm flux indices were downloaded from:

ftp://ftp.ngdc.noaa.gov/STP/space-weather/solar-data/solar-features/solar-radio/noontime-flux/penticton/penticton_observed/listings/listing_drao_noontime-flux-observed_monthly.txt.

## Appendix A: Sign flipping

The depicted PCs carry an arbitrary sign, that comes from the nature of the EOF analysis method. As we, for M2, perform an

EOF analysis independently at each altitude level, the sign of the EOFs and PCs can make sudden changes between the altitude layers. It does not affect the reconstruction, as the sign is simply canceled out in a reconstruction, but it results in a hashed image when visualized as in Fig. 5. To avoid this, we take the following steps.

First, the top altitude layer is chosen as first basis for the PC direction, $PC_b$ ($b$ denotes "basis"). Then, the following steps are performed, for each altitude layer, in descending order:

1. If $PC_b$ correlates negatively to the PC at the altitude level $i$, $PC_i$, the signs of $EOF_i$ and the $PC_i$ are both flipped (multiplied by $-1$).

2. The $PC_b$ is updated with the resulting $PC_i$, according to

$$PC_b \leftarrow (1-\alpha) \cdot PC_b + \alpha \cdot PC_i \qquad (A1)$$

and used as basis for the next altitude layer.





The factor $\alpha$ can take values between 0 and 1. $\alpha = 0.3$ seem to be working fine for us. $PC_b$ is updated no matter if a sign flipping takes place or not. It is therefore only used for holding information about the previous PCs, with fading influence with distance to the previous altitude layer.

5    It is done for all the principle components in the resulting data set. It creates a much smoother result, without sudden sign changes.

*Author contributions.* H. Wilhelmsen performed the computational implementation and the analysis, created the figures, and wrote the first draft of the paper. All authors contributed to the study design. F. Ladstädter provided advice to the analysis design and main contributions to the paper text. B. Scherllin-Pirscher provided advice on the method, on interpretation of the results and contributed to the paper text. A. K. Steiner provided guidance on all aspects of the study and contributed to the paper text and advised this work.

10    *Competing interests.* The authors declare that they have no conflict of interest.

*Acknowledgements.* We are grateful to UCAR/CDAAC (Boulder, CO, USA) for the provision of its RO phase and orbit data and to ECMWF (Reading, UK) for providing access to analysis and forecast data. The WEGC processing team members (M. Schwärz, B. Angerer) are thanked for providing the OPSv5.6 RO data. We acknowledge FU Berlin (Berlin, DE) for providing QBO wind indices, CPC for ENSO indices, and NOAA for the solar flux indices. We thank H. Rieder (WEGC, AT), T. Mendlik (WEGC, AT) and A. de la Torre (Universidad
15    Austral, AR) for their help and fruitful discussion. This work was funded by the Austrian Science Fund (FWF) under research grants P27724-NBL (VERTICLIM) and W1256-G15 (Doctoral Programme Climate Change – Uncertainties, Thresholds and Coping Strategies).



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



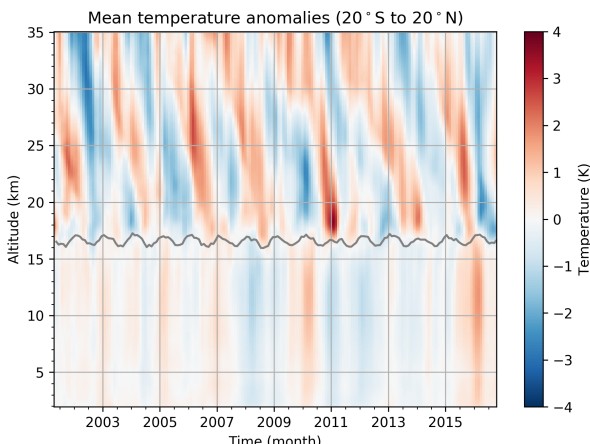

**Figure 1.** Zonal monthly mean temperature anomalies from GNSS RO from 20° S to 20° N and 2 km to 35 km. The gray line near 17 km indicates the tropopause height. Besides visualizing the features of the QBO in the RO record it can also be made audible through sonification. The interested reader can listen to QBO sounds under https://sysson.iem.at/sounds.html.







**Figure 2.** The first four EOFs computed from M1, EOF1 to EOF4 (left to right), are shown for selected altitudes at 30 km, 27 km, 22 km, 20 km, 17 km, 14 km, 11 km (top to bottom). The explained variance ratio is given in brackets in the titles.





**Figure 3.** The first four PCs computed from M1, PC1 to PC4 (top to bottom). The explained variance ratio is given in brackets in the titles.



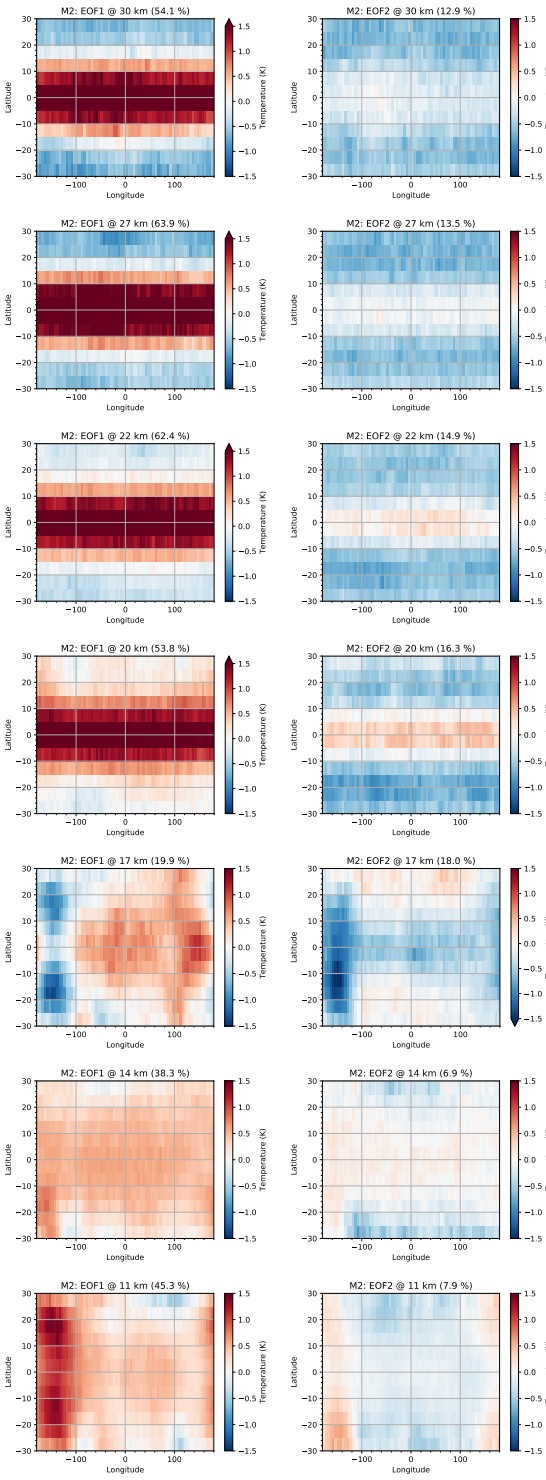

**Figure 4.** The first EOFs (left) and second EOFs (right) computed from M2, shown for selected altitude levels from the stratosphere (top) to the troposphere (bottom). The same altitude levels as for Fig. 2 are shown. The explained variance ratio is shown in brackets in the titles.



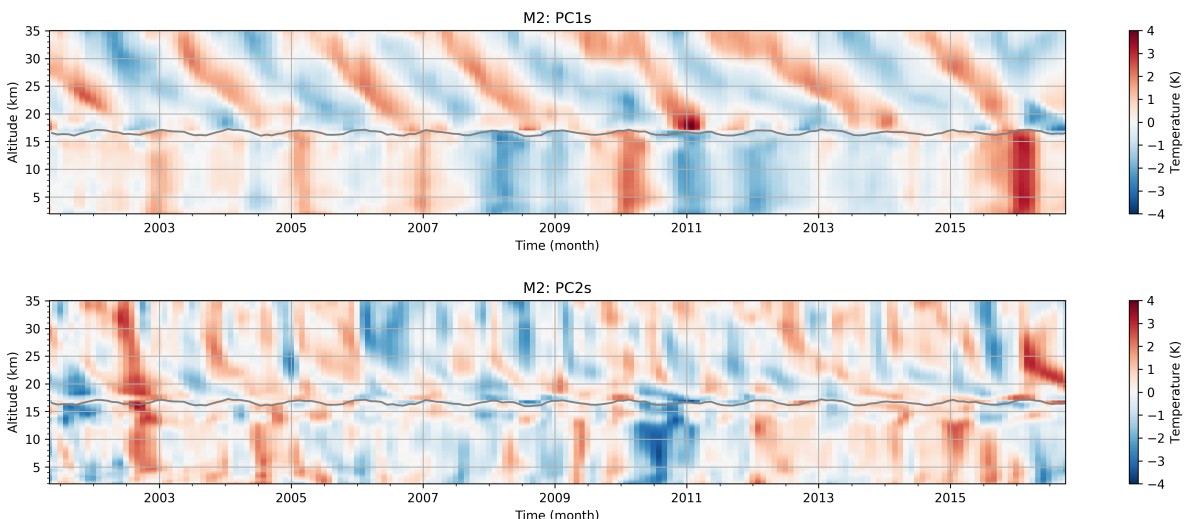

**Figure 5.** PC1s (top) and PC2s (bottom) from M2 for each altitude level. The gray line indicates the height of the tropopause.





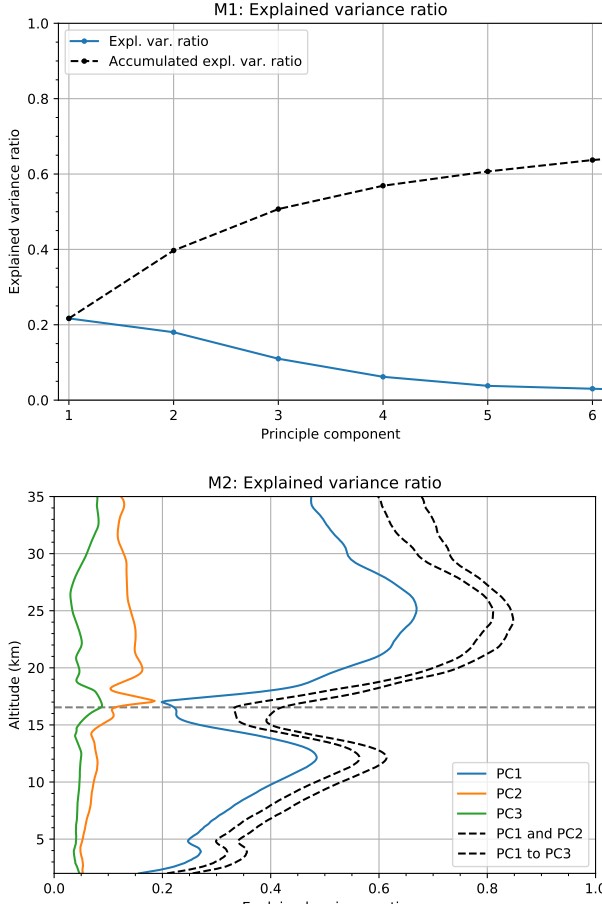

**Figure 6.** Explained variance ratio for M1 (top) shown as classical scree plot. Explained variance ratio for M2 (bottom) shown as function of altitude for PC1s to PC3s. The dashed lines show the cumulative sums of the explained variance ratios. The horizontal dashed gray line indicates the mean tropopause height.




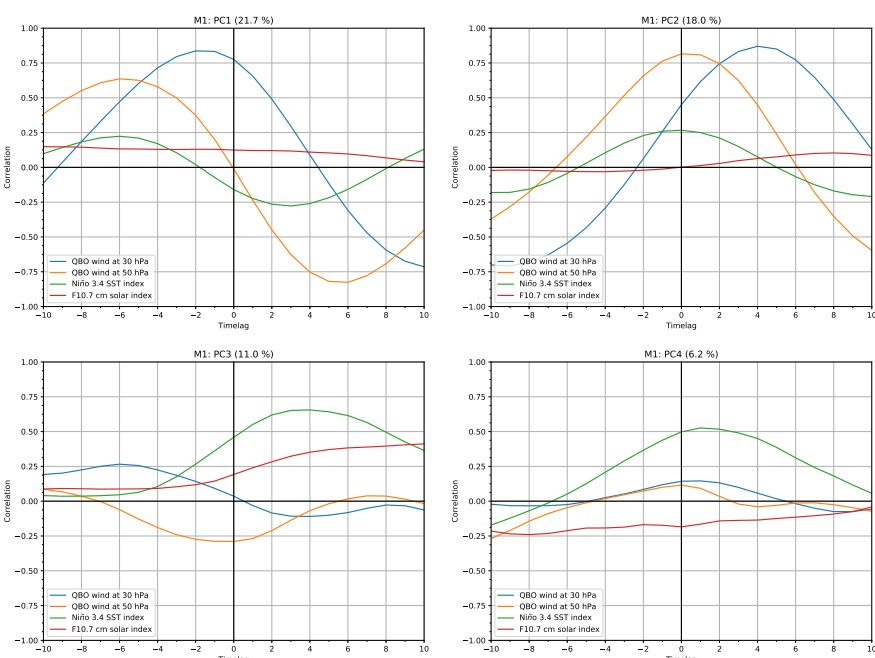

**Figure 7.** Correlations between derived indices from M1, PC1 to PC4, with conventional variability indices, QBO30, QBO50, Niño 3.4 SST, and F10.7 solar flux, shown for $\pm 10\ \mathrm{months}$ time lag. The explained variance ratio is given in brackets in the titles.




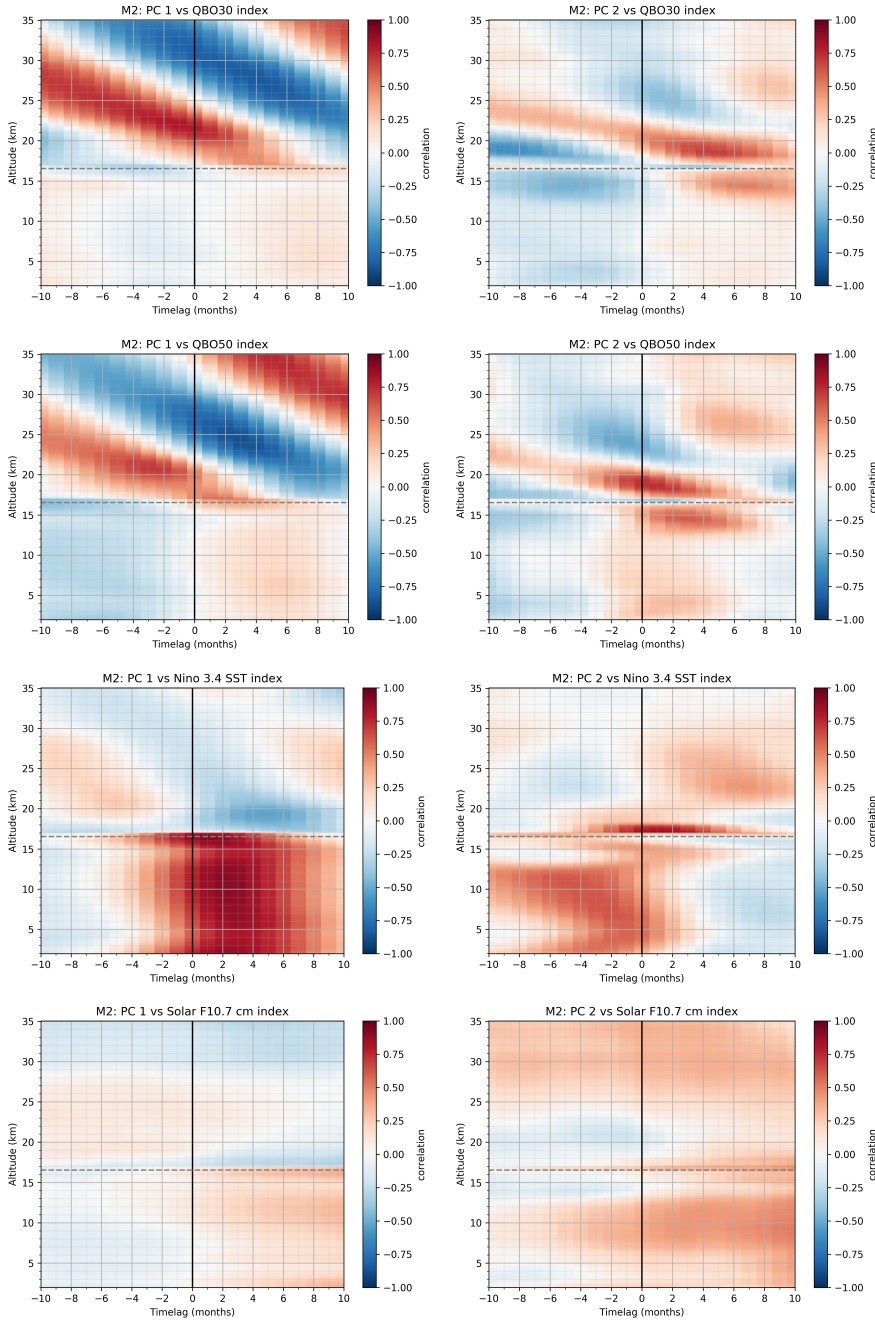

**Figure 8.** Correlations between the first PCs (left) and the second PCs (right) derived from M2 with known variability indices QBO30, QBO50, Niño 3.4, F10.7 solar flux (top to bottom), shown for each altitude level. The horizontal dashed line indicates the mean tropopause height.





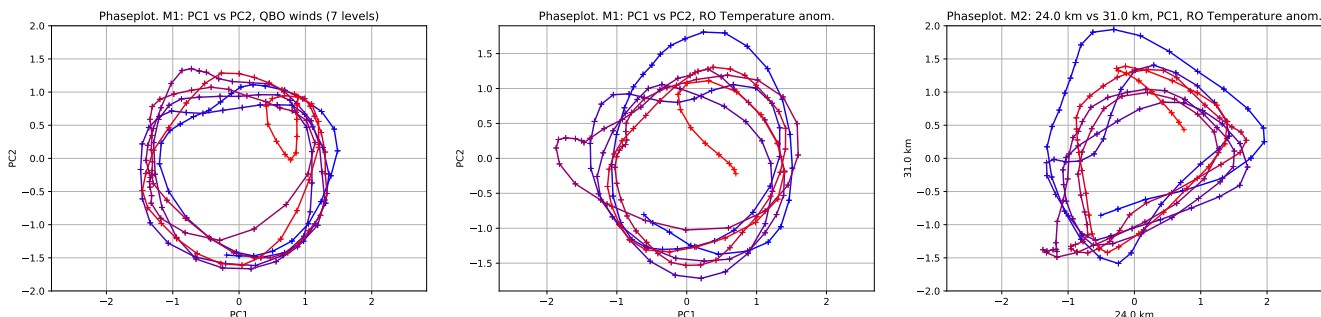

**Figure 9.** Phase space diagrams are shown for the RO time period May 2001 to October 2016. Blue color denotes the beginning of the period and turns into red towards the end of the period. PC1 vs PC2 from M1 based on QBO winds (left). PC1 vs PC2 from M1 based on RO temperature (middle), and PC1 at 24 km vs PC1 at 31 km from M2 based on RO temperature (right).




**Figure 10.** Reconstructed temperature fields from the first principal components which explain maximum variability. For M1 (left panels), reconstructed field using PC1 and PC2 (top), PC3 and PC4 (middle), and using PC1 to PC4 (bottom). For M2 (right panels), reconstructed field using the altitude resolved PC1s (top), using the altitude resolved PC2s (middle), and using PC1s plus PC2s (bottom). The gray line near 17 km indicates the tropopause height.



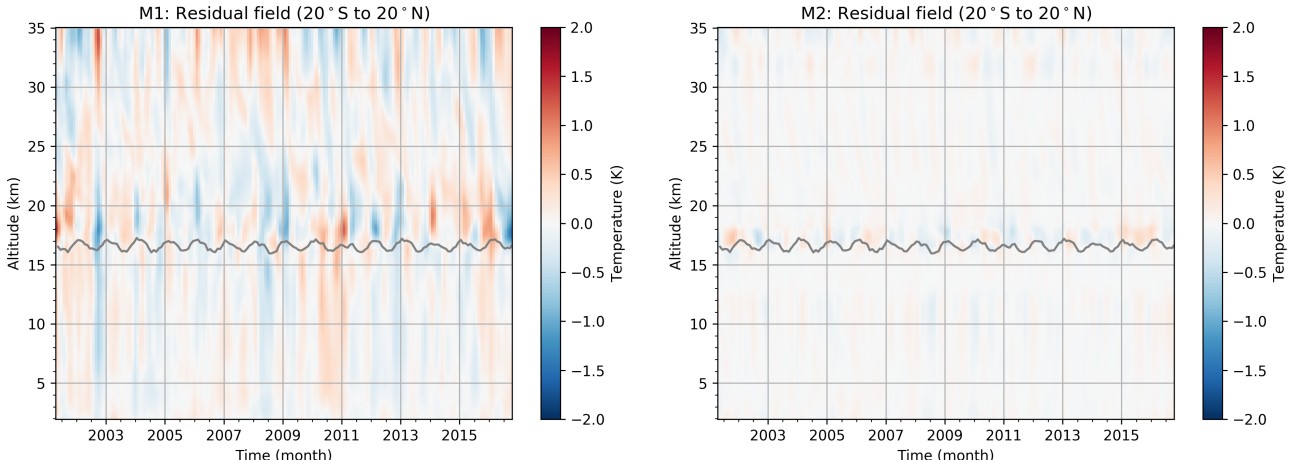

**Figure 11.** The residual temperature field for M1 (left) and M2 (right) showing the difference of the input minus the reconstructed fields, using PC1 to PC4 for M1 and PC1s to PC2s for M2. The gray line near 17 km indicates the tropopause height.