# Peer review of "Atmospheric QBO and ENSO indices with high vertical resolution from GNSS radio occultation temperature measurements"

_Atmospheric Measurement Techniques, 2017_

## Referee Comment (RC5)

Review:

Atmospheric QBO and ENSO indices with high vertical resolution from GNSS
radio occultation temperature measurements
by Wilhelmsen et al.

This paper utilizes GNSS RO temperature measurements to build a new
atmospheric QBO and ENSO indices by applying two different kinds of EOF
analysis.  Those indices, independent from the conventional ones, will be useful
in climate variability study.  I suggest a couple minor suggestions.

1. The paper describes the EOF method in detail, however, does not provide
   enough justification of using this method and what is the benefits of this
   method, compared to the conventional anomaly analysis.  For example,
   normalized temperature anomaly time series at each level also can
   represent the relative strength of the temperature variation at a given
   level, similar to that shown with EOFs.  A few reasons can be included:

    1) The EOF analysis can extract the major modes from small scale
   spacial and temporal variations.  This is an important aspect of the EOF
   method, which can screen out sampling errors from irregular sampled
   data, i.e., GNSS RO observations.
   2) The EOF analysis can be used to explore the structure of the variability
   within a data set in an objective way, and to analyze relationships within a
   set of variables.  In this study, EOF2 method account the temperature at
   each level as independent set of variables and connect their modes to
   analyze the vertical structures of the temperature variation.
2. The meaning of eigenvectors can be explained to further use them in the
   reconstruction of the anomaly field.
3. The paper shows the linear correlations between PCs and ENSO, QBO,
   and F10.7 indices, respectively.  I recommend to show those conventional
   indices along with newly suggested PCs from M2 at given levels (e.g.,
   surface, 50hPa) in the your PC plots.   It will be an effective way to
   demonstrate the similarities and differences of new indices.

---

## Referee Comment (RC1) · Anonymous Referee #1 · 11 Aug 2017

The main purpose of the paper is to demonstrate that the GNSS data can be used for studies of tropical atmospheric variability in both the stratosphere and the troposphere. Using tropical temperatures from GNSS measurements the authors produce QBO and ENSO indices by several versions of principal component analysis. These indices are then discussed and compared to traditional indices.

I find that the paper contains some interesting results and that it is relatively well written. I have a few major points that the authors should address before I can recommend that the paper is accepted.

Major comments:

1) The authors stress the high vertical resolution of their data (line 2 in abstract and line 17 in section 5). However, I don't think it is demonstrated anywhere in the paper that this is important. Actually, in section 4.1 it is mentioned that the patterns from M1 are not sensitive to the vertical resolution.

Would the same results be obtained if the analyses were performed with re-analysis data (e.g., NCEP or ERA)? If this is the case – and I think the authors should check – then the importance of the GNSS data may not be so high.

2) In the principal component analysis the authors include the latitude information. Normally when the QBO is studied from the winds latitudinal means are used. What is the reason for not using latitudinal means in this study? Does it make any difference?

3) Can anything be said about the coupling of the ENSO and the QBO? The method M1 includes all levels both in the troposphere and in the stratosphere so I wonder if it would be possible to gain insight into the proposed connection between these two parts of the atmosphere.

Minor comments:

Page2, l5: The new paper by Dunkerton (10.1002/2017JD026542) could be included here.

Page2, l18: The paper by Christiansen et al. (10.1002/2016GL070751) suggesting a coupling between ENSO and QBO could be cited here.

Page2, l26: This sentence is unclear.

Section 2, l26: How many grid-points with missing data do you have? " .. boundaries ..": But the data-set is global?

Page 3, top: How much does this prior knowledge influence the temperature in the tropopause? Is it a large part or can the ENSO be seen in the raw GNSS data alone?

Section 3.1, line 4: Does the centering matter? Is this not already included when you

use the covariances?

Page 6, l4: Antipodal? Is this the right word?

Page 6, l8-13. I found it hard to follow this. What does "this pattern" refer to? Is there a QBO pattern in the stratosphere with a longitudinal structure?

It might be a good idea to merge section 4.6 with sections 4.1 ad 4.2. Many of the questions that arise reading sections 4.1 and 4.2 are answered in section 4.6

Fig. 1: How is the tropopause calculated?

Fig. 2: Perhaps the standard pressures corresponding to these vertical levels could be given.

Perhaps the last sentence in the abstract and the sentence in section 5 beginning with "We provide .." should be removed. They sound as if you want to sell me a used car.

---

## Referee Comment (RC2) · Anonymous Referee #2 · 1 Sep 2017

The authors present two new methods in order to analyze the 4-D field of global temperature variability. They show that the QBO and ENSO signals are also visible in the temperature. The two new applications of EOF and PC analysis are clearly presented.

I am a bit skeptical if the new analysis technique really leads to new knowledge and understanding of atmospheric variability. In case of QBO and ENSO the informations of the wind changes and the water vapor loading of the troposphere might be more direct than the temperature information.

In the stratosphere, the temperature-QBO might be related to the QBO in vertical wind. It would be nice if the authors would explain the physical relationships between the

temperature anomalies and other anomalies. What are the basic reasons for the occurrence of the temperature anomalies? Generally I recommend the publication of this article which might be a good inspiration for future studies on atmospheric variability.

Minor comments: 1) Introduction: I am missing the discussion of existing literature about the ENSO effects in the stratosphere. Do you see such an effect in your ENSO indices at stratospheric altitudes?

p.3 line 24 ... from from May 2001 ....

---

## Referee Comment (RC3) · Anonymous Referee #3 · 12 Sep 2017

This paper presents the results of Empirical Orthogonal Function (EOF) analysis applied to temperature data obtained from the Global Navigational Satellite system (GNSS). Others have detected a QBO signal in GNSS temperatures, and others have done an EOF analysis of QBO winds. But the current work applies EOF analysis to GNSS temperatures, and this represents a new and original contribution.

This paper demonstrates that the GNSS data provide a straightforward means of obtaining and assessing the current and past states of both the QBO and ENSO. In another journal one might expect a deeper discussion of the dynamical significance of the signals shown, but for Atmospheric Measurement Techniques the level of this

discussion is acceptable as it stands. The emphasis of the paper is quite properly on the technique and the usefulness of its resulting indices.

I therefore recommend that this paper be accepted for publication with some minor revisions. There are some aspects that could be improved, especially with regard to its clarity and to improve its overall effectiveness. Specific points are listed below:

Scientific points:

p. 6, line 14. EOF3 and EOF3 resemble EOF1 and EOF2 just above the tropopause. Does this imply that ENSO is modulating the QBO in some way? Or is this some kind of numerical leakage, with no more physical meaning than the flipping of M2 PCs between QBO and ENSO dominance at different altitudes?

p. 6, line 26-27. The authors comment on the top half of Figure 5, where the QBO and ENSO patterns are clearly visible above and below the tropopause, respectively. But what does the bottom half of Figure 5 tell us? There are hints of a QBO-like propagation in the lower stratosphere. But other than that it is unclear how the M2 PC2s should be interpreted.

p. 6, line 25; and p. 8, lines 17-34: It might be helpful to the casual reader to explain more fully how Figure 5 differs from the right-hand upper and middle panels of Figure 10. The differences between plotting the M2 PC time series, and the M2 time series reconstituted from the PCs, might not be entirely clear at first glance.

p. 9, line 30: It is not surprising that the second method captures more of the variability. If you think of the two analyses as being akin to different kinds of statistical curve-fitting, there are a great many more coefficients in M2 to which the "fit" is being made. Smaller residual variances will naturally follow. The key question here is, is there a physical meaning to the increased fits? I suspect the answer lies in the clear relationship between signals at different altitudes. Perhaps computing time series coherence between altitudes would show formally what the eye can clearly see.

p. 9, lines 25-27: The lack of a known time lag in the M1 method is alluded to on p. 7 in the discussion on Figure 7. But perhaps a little more discussion of this could be added in the method discussion in Section 3.1.

Grammatical/stylistic points:

The formatting of the spatial points if preparing for the EOF analysis is referred to in two different ways: as "stringing along a single axis" (p. 4 line 22) and "reshaping" (p. 5, line 28). The term "reshaping" seems preferable. If "stringing" is retained, please note that the past tense of "string" is "strung", not "stringed".

p. 2, line 10: "origins" should be "originates"

p. 6, line 20: "more coarse" should be "coarser"

Figure 1. If the Nino 3.4 SST index could be plotted along the bottom of this Figure, and the QBO30 and QBO50 winds plotted at their respective altitudes, it would establish for the reader early on the relationship between these traditional indices and the measured temperature field. These would not have to be quantitative plots with overlaid labelled axes; simple unlabeled time series, similar to the tropopause altitude in gray, would suffice. Granted, the tropopause curve needs no labeling, since it varies along the labelled y-axis. But showing how the original temperatures relate to these indices would be helpful preparation for what follows in the paper.

Figures 2, 4, and 8: The use of small-multiple plots here is good. But instead of simple pasting together independent plots, each with its own title and color bar, these Figures would be improved by inserting the small plots into a labelled grid structure. For example, in Figure 2, the altitudes should be clearly labelled along the left-hand side, by each corresponding row of plots. Likewise, the EOF numbers should be indicated along the top of the figure, at the top of each column of plots. The explained variance could be retained in each plot's title, but moving the other title information to the grid margins would greatly improve readability. And eliminating all but one color bar would

make it instantly clear that the scale is not changing from plot to plot.

Figure 6. This Figure would be improved if the x- and y-axes in the top half were to be exchanged, so that the x-axis on the top figure matched that of the bottom, making them easier to compare.

[Figure]

---

## Author Response (AR1)

**List of relevant changes**

**Manuscript: amt-2017-226**

**Title:** "Atmospheric QBO and ENSO indices with high vertical resolution from GNSS radio occultation temperature measurements"

**Authors:** Hallgeir Wilhelmsen, Florian Ladstädter, Barbara Scherllin-Pirscher, and Andrea K. Steiner.

Please refer to the attached responses to the referees for changes in the manuscript. Note that the page and line numbers refer to the manuscript uploaded on July 13, 2017 (amt-2017-226-manuscript-version2.pdf).

In addition to these changes, the following relevant changes have been made to the revised manuscript:

1. The data used in the first manuscript had the time period from May 2001 to October 2016, with 100 meter vertical resolution. In the revised manuscript, all calculations have been done again, with an updated time period from September 2001 to February 2017, with 200 meter vertical resolution. These changes have no implications on the conclusions of this study. All figures and numbers have been updated accordingly in the revised manuscript.

2. We added two new subfigures to Fig. 7 to better point out the relationship between PC3 and PC4 with ENSO, updated the caption, and added a description in the manuscript (Sect. 4.4).

3. We changed the altitudes used in the rightmost subplot of Fig. 9, to better compare to the other phase plots.

[Figure]

AMTD

Interactive
comment

We thank the reviewer for the positive comments and the constructive questions. Please find our response below.

[Figure]

Major comments

**Comment 1a:** "The authors stress the high vertical resolution of their data (line 2 in abstract and line 17 in section 5). However, I don't think it is demonstrated anywhere in the paper that this is important. Actually, in section 4.1 it is mentioned that the patterns from M1 are not sensitive to the vertical resolution."

**Response 1a:** The main idea of the presented study is based on introducing and demonstrating that M2 is able to exploit the vertical resolution of RO. This is mentioned several times in the manuscript, e.g., page 1, line 7 – line 9, or page 9, line 24 – line 27.

It is true that the patterns from M1 are not sensitive to the vertical resolution of the input data as stated in Sect. 4.1.

With M2, however, we extract the main atmospheric variability modes at each altitude level. Therefore, variability modes have the same vertical resolution as the input data set.

We get atmospheric variability indices with high vertical resolution *because* we use input data with high vertical resolution. For the method M2 itself, the high vertical resolution is not important.

**Comment 1b:** "Would the same results be obtained if the analyses were performed with re-analysis data (e.g., NCEP or ERA)? If this is the case – and I think the authors should check – then the importance of the GNSS data may not be so high."

**Response 1b:** We expect that both M1 and M2 would yield similar results when using reanalysis data sets instead of RO as input field. However, differences between reanalysis and the RO observational data set can be substantial, especially in the tropopause region and above. In Fig. 1 in this response we show differences between the monthly mean temperature fields from RO and ERA-I, which assimilates RO.

[Figure]

The distinct differences in the tropopause region and the stratosphere as well as the QBO-like signatures in the difference, stem from a known bias of ERA-I (Poli et al. 2010, https://doi.org/10.1002/qj.722, and S. Healy, personal communication).

Also, e.g. Long et al., (2017, https://doi.org/10.5194/acp-2017-289) states that all existing reanalysis data sets have difficulties describing the QBO winds.

We therefore expect that our description of the variability, especially in regions where RO is known to be of best quality, is more accurate.

**Comment 2:** "In the principal component analysis the authors include the latitude information. Normally when the QBO is studied from the winds latitudinal means are used. What is the reason for not using latitudinal means in this study? Does it make any difference?"

**Response 2:** We assume that the reviewer's question refers to using zonal means in the analysis.

As can be seen in Fig. 4 in the manuscript, the EOF patterns reveal both latitudinal and longitudinal patterns, depending on the altitude. There is only minor longitudinal variation in the stratosphere, where the QBO is the dominant variability pattern. However, in the tropopause region and below, there is a distinct longitudinal variation.

To analyze the impact of the longitudinal variability on our results, we repeated the analysis for M2, using only zonal means from RO temperature. In Fig. 2 in this response we show the difference of the resulting PCs from using zonal means to the PCs including the latitudinal information as shown in the manuscript (Fig. 5). We find only minor differences in the stratosphere but distinct differences near the tropopause and in the troposphere.

The main goal of the methods is to capture the atmospheric variability at the respective altitude levels, and not only the variability originating from the QBO. This is the reason why we do not use latitudinal bands in our analysis, but include the whole field.

[Figure]

**Comment 3:** "Can anything be said about the coupling of the ENSO and the QBO? The method M1 includes all levels both in the troposphere and in the stratosphere so I wonder if it would be possible to gain insight into the proposed connection between these two parts of the atmosphere."

**Response 3:** We thank the reviewer for pointing to this interesting topic. In the revised version of the manuscript we cite several additional studies where the coupling of ENSO and QBO and teleconnections are discussed.

As the reviewer suggests, method M1 might be useful to investigate connections between the troposphere and the stratosphere.

However, the main focus of this paper is to describe a method to detect the atmospheric variability in the QBO and ENSO regions. Investigating coupling effects is beyond the scope of this work and would require a dedicated study.

We added the following paragraph in the introduction:

"The interaction between ENSO and QBO has been investigated in many studies (Taguchi, 2010; Schirber, 2015; Christiansen et al., 2016; Geller et al., 2016; Hansen et al., 2016). For further literature on the relationship between ENSO, QBO, and teleconnections, see e.g. the introduction in Dunkerton (2017) and references within."

Minor comments

**Comment 4:** Page 2, line 5: "The new paper by Dunkerton (10.1002/2017JD026542) could be included here."

**Response 4:** Added to page 2, line 5.

**Comment 5:** Page 2, line 18: "The paper by Christiansen et al. (10.1002/2016GL070751) suggesting a coupling between ENSO and QBO could be cited here."

[Figure]

**Response 5:** We added the reference in the introduction section.

**Comment 6:** Page 2, line 26: "This sentence is unclear."

**Response 6:** We replaced the sentence

"QBO characteristics can be exploited in the commonly used QBO indices, often derived from wind speeds, serving as proxies to describe the QBO."

with

"The QBO is often characterized by wind measurements."

**Comment 7:** Section 2, line 26: "How many grid-points with missing data do you have? "..boundaries..": But the data-set is global?"

**Response 7:** We thank the reviewer for pointing to this question.

The EOF analysis method requires that there are no missing data. We did the calculations on a 30°S to 30°N slice of the global data set. After doing the bilinear interpolation, missing numbers were still found at the boundaries of the 30° latitude limits.

In the revised manuscript, we first do the bilinear interpolation on the global grid. We then select the ±30° latitudinal band. The term "boundaries" is therefore not needed any more.

The number of missing points depends on altitude and time. In the beginning of the time series (the first 6 years), about 10 % to 30 % of the 5° x 5° latitude-longitude grid do not contain data. In the worst case (only in the first month of the time series), up to 60 % of the data is missing. Later, starting 2006, when more RO missions contribute, there are no missing data in the investigated spatial domain, 30°S to 30°N.

We replaced

"Grid points with missing data are filled horizontally using bilinear interpolation, at each time step, or filled with the nearest neighbor data at the boundaries."

[Figure]

with

"At each time step and each altitude level, grid points with missing data are filled horizontally using bilinear interpolation."

**Comment 8:** Page 3, top: "How much does this prior knowledge influence the temperature in the tropopause? Is it a large part or can the ENSO be seen in the raw GNSS data alone?"

**Response 8:** Up to 25 km the impact of the high-altitude initialization on the RO temperature is small. Therefore the tropopause is not influenced by the prior knowledge. See Angerer et al. (in press, 2017, https://doi.org/10.5194/amt-2017-225).

**Comment 9:** Section 3.1, line 4: "Does the centering matter? Is this not already included when you use the covariances?"

**Response 9:** Yes, the centering is already included.

We replaced the sentence, page 4, line 22:

"The resulting matrix is therefore a two dimensional matrix, in space ($s$) and time ($t$) only, $X(s_{(\phi,\theta,z)}, t)$, represented by Eq. (1)."

with

"The resulting matrix is therefore two-dimensional, in space ($s$) and time ($t$) only, $X(s_{(\phi,\theta,z)}, t)$, represented by Eq. (1), where each row, $\vec{x}_{s_p}(t)$, corresponds to a time series at a specific location (in $\phi, \theta, z$)."

and removed the whole sentence, page 5, line 4:

"Each row, $\vec{x}_{s_p}(t)$, which represents a time series at a specific location, is *centralized* by subtracting the arithmetic mean of the time series at each grid point."

**Comment 10:** Page 6, line 4: "Antipodal? Is this the right word?"

**Response 10:** We replaced "antipodal" with "with opposite sign".

[Figure]

**Comment 11:** Page 6, line 8 – 13: "I found it hard to follow this. What does "this pattern" refer to? Is there a QBO pattern in the stratosphere with a longitudinal structure? It might be a good idea to merge section 4.6 with sections 4.1 ad 4.2. Many of the questions that arise reading sections 4.1 and 4.2 are answered in section 4.6."

**Response 11:** We agree that this sentence is not clear, and changed it (see below). We considered merging Sect. 4.6 with Sect. 4.1 and Sect. 4.2, but we prefer the current structure.

We replaced

"This pattern around the tropopause is also visible in the third and the fourth EOFs (EOF3 and EOF4 respectively)."

with

"This longitudinal variability pattern around the tropopause is also visible in the third and the fourth EOF (EOF3 and EOF4 respectively)."

**Comment 12:** "Fig. 1: How is the tropopause calculated?"

**Response 12:** The tropopause height is calculated according to the WMO definition of the lapse rate tropopause (WMO, 1957) on the monthly mean temperature profiles. Further information on the algorithm we use can be found in Rieckh et al. (2014, http://dx.doi.org/10.5194/amt-7-3947-2014).

We replaced the line in the figure caption

"The gray line near 17 km indicates the tropopause height."

with

"The gray line near 17 km indicates the tropopause height for the monthly mean temperature profiles, calculated according to the WMO definition of the lapse rate tropopause (WMO, 1957)."

[Figure]

**Comment 13:** "Fig. 2: Perhaps the standard pressures corresponding to these vertical levels could be given."

**Response 13:** We added this information to Fig. 2 in the revised manuscript.

**Comment 14:** "Perhaps the last sentence in the abstract and the sentence in section 5 beginning with "We provide .." should be removed. They sound as if you want to sell me a used car."

**Response 14:** We reformulated the respective sentence in the summary section from

"We provide vertically high resolved atmospheric variability indices which can deliver improved information on the natural variability patterns such as QBO and ENSO."

to

"Vertically high resolved atmospheric variability indices can deliver improved information on the natural variability patterns such as QBO and ENSO."
* * *
[Figure]

[Figure]

**Fig. 1.** Monthly mean temperature differences from RO and ERA-I.

[Figure]

[Figure]

**Fig. 2.** Difference of PC1 and PC2 obtained from M2 using zonal means and M2 using the whole input field.

[Figure]

Atmos. Meas. Tech. Discuss.,
doi:10.5194/amt-2017-226-AC2, 2017

[Figure]
We thank the reviewer for helpful questions and comments. Please find our responses below.

[Figure]

Major comments

**Comment 1:** "In the stratosphere, the temperature-QBO might be related to the QBO in vertical wind. It would be nice if the authors would explain the physical relationships between the temperature anomalies and other anomalies.

What are the basic reasons for the occurrence of the temperature anomalies?"

**Response 1:** The main connection between QBO winds and temperature, $T$, is expressed through Eq. (1),

$$\frac{\partial \vec{u}}{\partial z} = -\frac{R}{H\beta}\frac{\partial^2 T}{\partial y^2},\tag{1}$$

where $\vec{u}$ is the zonal wind speed, $z$ is the log-pressure height, $R$ is the gas constant, $H$ the nominal scale height, $\beta$ is the latitudinal derivative of the Coriolis parameter, and $y$ is latitude.

Centered on the equator, with meridional scale $L$, Eq. (1) can be approximated as

$$\frac{d\vec{u}}{dz} \sim \frac{R}{H\beta}\frac{T}{L^2}.\tag{2}$$

The relationship between the zonal winds and the temperature anomalies around the equator is therefore proportional to the vertical gradient of the zonal winds. See e.g., Randel et al. (1999, https://doi.org/10.1175/1520-0469(1999)056<0457:GQCDFU>2.0.CO;2), or Baldwin et al. (2001, https://doi.org/10.1029/1999RG000073).

We added to page 2, line 3:

"The atmospheric temperature anomalies are proportional to the vertical gradient of the zonal winds (Randel et al., 2001; Baldwin et al., 2001), which makes it possible to investigate the QBO through temperature anomalies."

[Figure]

Minor comments:

**Comment 2:** "Introduction: I am missing the discussion of existing literature about the ENSO effects in the stratosphere. Do you see such an effect in your ENSO indices at stratospheric altitudes?"

**Response 2:** We added several references to the introduction, see response to Comment 3 from Referee #1, RC1, and have some additional discussion about this in response to Comment 1 from Referee #3, RC3.

**Comment 3:** Page 3, line 24: "... from from May 2001".

**Response 3:** One "from" removed.

[Figure]

Atmos. Meas. Tech. Discuss.,
doi:10.5194/amt-2017-226-AC3, 2017

[Figure]

We thank the reviewer for valuable questions and comments helping to improve this paper. They are addressed in the responses below.

[Figure]

Major comments

**Comment 1:** Page 6, line 14: "EOF3 and EOF3 resemble EOF1 and EOF2 just above the tropopause. Does this imply that ENSO is modulating the QBO in some way? Or is this some kind of numerical leakage, with no more physical meaning than the flipping of M2 PCs between QBO and ENSO dominance at different altitudes?"

**Response 1:** It is possible that the ENSO signal propagates through the tropopause, modulating the QBO in the lower stratosphere. There are several studies related to this topic. However, investigating this interesting subject is beyond the scope of this work.

We cite several additional studies in the revised version of the manuscript, discussing these topics.

We think that the signal seen in EOF3 at 20 km (Fig. 2) could be related to ENSO, while the signals in EOF3 and EOF4 further above in the stratosphere are probably due to numerical leakage.

Please see response to Comment 3 from Referee #1, RC1.

We added a paragraph in the introduction:

"The interaction between ENSO and QBO has been investigated in many studies (Taguchi, 2010; Schirber, 2015; Christiansen et al., 2016; Geller et al., 2016; Hansen et al., 2016). It is, however, beyond the scope of this work and would require a dedicated study. For further literature on the relationship between ENSO, QBO, and teleconnections, see e.g. the introduction in Dunkerton (2017) and references within."

**Comment 2:** Page 6, line 26 – 27. "The authors comment on the top half of Figure 5, where the QBO and ENSO patterns are clearly visible above and below the tropopause, respectively. But what does the bottom half of Figure 5 tell us? There are hints of a QBO-like propagation in the lower stratosphere. But other than that it is unclear how the M2 PC2s should be interpreted."

[Figure]

**Response 2:** We agree that a discussion about the lower panel of Fig. 5 is missing. In the revised version of the manuscript we make the following changes to Sect. 4.2.

We changed page 6, line 25 from

"The first set of PCs reveals a separation into a part above the tropopause and into a part below the tropopause"

to

"As for the EOFs, each PC represents an altitude level.

The first set of PCs (upper panel of Fig. 5) reveals a separation into a part above the tropopause and into a part below the tropopause."

and added the following at the end of Sect. 4.2:

"The separation is not as clear in the second set of PCs (lower panel of Fig. 5). It shows part of the residual variability that is not described by the first set of PCs. These also show downward propagating patterns, but different from the first set of PCs, and less regular in the stratosphere.

Keep in mind that both the EOFs and the PCs have been scaled by their corresponding eigenvalues (see Sect. 3.1), at each altitude level separately. The corresponding eigenvalues are proportional to the explained variance ratios (see Fig. 6). The magnitude of each time series does therefore not necessarily describe its importance, nor are the contributions from each EOF or PC directly comparable. The actual contribution can be seen in the reconstruction. See Sect. 4.3 and Sect. 4.6 for further details.

Also keep in mind that dominating atmospheric variability at different altitude levels can be caused by different physical mechanisms. The physical context of the first principal components may therefore also change with altitude because the calculations are performed separately at each altitude level for M2. Finally, if independent variability modes explain a similar amount of variance, their corresponding PC time series can

switch between two PCs (e.g., between PC1 and PC2) at neighboring altitude levels."

**Comment 3:** Page 6, line 25 and page 8, lines 17 – 34: "It might be helpful to the casual reader to explain more fully how Figure 5 differs from the right-hand upper and middle panels of Figure 10. The differences between plotting the M2 PC time series, and the M2 time series reconstituted from the PCs, might not be entirely clear at first glance."

**Response 3:** For the Page 6, line 25, see Response 2 above.

For page 8, lines 17 – 34, we inserted the following sentence:

"In contrast to Fig. 5, where the PC1s and the PC2s are plotted in EOF space, we map the PC1s and the PC2s back into anomaly space."

We also changed the paragraph starting at page 8, line 26 from

"Figure 10 (bottom left panel) shows a combination of all four principle components from M1, and the result well resembles the input temperature anomalies shown in Fig. 1."

to

"Figure 10 (bottom left panel) shows the reconstruction using all four principal components from M1. This time series well resembles the input temperature anomalies shown in Fig. 1."

**Comment 4:** Page 9, line 30: "It is not surprising that the second method captures more of the variability. If you think of the two analyses as being akin to different kinds of statistical curve-fitting, there are a great many more coefficients in M2 to which the "fit" is being made. Smaller residual variances will naturally follow. The key question here is, is there a physical meaning to the increased fits? I suspect the answer lies in the clear relationship between signals at different altitudes. Perhaps computing time series coherence between altitudes would show formally what the eye can clearly see."

[Figure]

**Response 4:** Figure 1 in this response shows the correlation between a specific PC (PC1 or PC2) from M2 (see Fig. 5 in the manuscript) at a given altitude level (e.g., at 17 km) and PC1/PC2 (also from M2) at all altitude levels. Correlation plots are shown for seven selected levels.

Figure 1 in this response reveals that these correlation patterns are very similar to correlation plots shown in Fig. 8 of the manuscript. The cross correlation (third and fourth column) is particularly high around the tropopause which could be related to the drop in explained variance ratio (Fig. 6 in the manuscript). This is subject to further investigation.

**Comment 5:** Page 9, lines 25 – 27: "The lack of a known time lag in the M1 method is alluded to on p. 7 in the discussion on Figure 7. But perhaps a little more discussion of this could be added in the method discussion in Section 3.1."

**Response 5:** We agree that it could help to add some information leading to Fig. 7 this early in the manuscript. Since the meaning of the time lag is discussed in Sect. 4.4, we think it suffices to lead towards physical interpretation of the indices in Sect. 3.1.

We changed page 5, line 11 from

"The first few PCs from Eq. (3) can now be used as proxies for the temporal variability, which we call *indices* in the following."

to

"The first few PCs from Eq. (3) can now be used as proxies for the temporal variability. We call these *indices* in the following. Calculating cross correlations between these indices and conventional indices can point to their physical interpretation."

[Figure]

Grammatical/stylistic points

**Comment 6:** "The formatting of the spatial points if preparing for the EOF analysis is referred to in two different ways: as "stringing along a single axis" (p. 4 line 22) and "reshaping" (p. 5, line 28). The term "reshaping" seems preferable. If "stringing" is retained, please note that the past tense of "string" is "strung", not "stringed"."

**Response 6:** We changed the sentence

"The space dimensions are all stringed along a single axis..."

into

"The space dimensions are all reshaped into a single axis"

and

"Therefore, only the latitude and longitude dimensions are stringed along one axis"

into

"Therefore, only the latitude and longitude dimensions are reshaped into one axis"

**Comment 7:** Page 2, line 10: ""origins" should be "originates"."

**Response 7:** Corrected.

**Comment 8:** Page 2, line 20: ""more coarse" should be "coarser"."

**Response 8:** Corrected.

**Comment 9:** "Figure 1. If the Nino 3.4 SST index could be plotted along the bottom of this Figure, and the QBO30 and QBO50 winds plotted at their respective altitudes, it would establish for the reader early on the relationship between these traditional indices and the measured temperature field. These would not have to be quantitative plots with overlaid labelled axes; simple unlabeled time series, similar to the tropopause

altitude in gray, would suffice. Granted, the tropopause curve needs no labeling, since it varies along the labelled y-axis. But showing how the original temperatures relate to these indices would be helpful preparation for what follows in the paper."

**Response 9:** We added an overlay, depicting the indices in Fig. 1, or adding panels above (for QBO) and below (for ENSO) depicting the QBO winds at selected pressure levels and the Niño 3.4 SST index, respectively.

**Comment 10:** "Figures 2, 4, and 8: The use of small-multiple plots here is good. But instead of simple pasting together independent plots, each with its own title and color bar, these Figures would be improved by inserting the small plots into a labelled grid structure. For example, in Figure 2, the altitudes should be clearly labelled along the left-hand side, by each corresponding row of plots. Likewise, the EOF numbers should be indicated along the top of the figure, at the top of each column of plots. The explained variance could be retained in each plot's title, but moving the other title information to the grid margins would greatly improve readability. And eliminating all but one color bar would make it instantly clear that the scale is not changing from plot to plot."

**Response 10:** Thanks for this valuable suggestions. We improved these figures according to the suggestions.

**Comment 11:** "Figure 6. This Figure would be improved if the x- and y-axes in the top half were to be exchanged, so that the x-axis on the top figure matched that of the bottom, making them easier to compare."

**Response 11:** Thanks also for this input. We improved Fig. 6 accordingly.

[Figure]

M2: Cross and auto correlations

[Figure]

**Fig. 1.** Cross- and auto correlations using PC1(z)s and PC2(z)s from M2. Each column represents the correlations for a reference PC(z) (black line) vs. the PC(z)s for another (or the same) principle component.

[Figure]

Atmos. Meas. Tech. Discuss.,
doi:10.5194/amt-2017-226-AC4, 2017

[Figure]

We would like to thank the reviewer for the helpful comments and questions. Please see our responses below.

**Comment 1:** "The authors applied two different methods in order to demonstrate the advantage of the high resolution of GNSS RO profiles. This result, according to lines 6-10 of page 9, seems to be inferred from what the authors called reconstructed fields. I'm not sure if PCA methodology allows calling reconstructed patterns by multiplying

[Figure]

the PC loadings by PC scores (see below). Please add some reference about it or explain this concept."

**Response 1:** We added a citation to page 8, line 15:

"We do this by multiplying the EOF with its corresponding PC (Wilks, 2006)."

**Comment 2:** "PCA results change according to the input matrix and they can be different considering for example, a domain between 60 and 60. I think the authors should show some result or make some comparison."

**Response 2:** The sensitivity of the different methods is an interesting topic that we only briefly discuss in the manuscript.

In this work we focus on tropical atmospheric temperature variability. The method is equally valid for other latitudinal regimes. We plan to create indices also for the mid and high latitude regions in future work.

**Comment 3:** "Perhaps calling PC loading fields to what authors called "EOF" and PC scores to the time series that they call "PC" it would be better, since it would agree with the common terminology for S-Mode in PCA (EOF)."

**Response 3:** We acknowledge that the labels can be confusing. As also discussed in the review paper on EOF analysis / PCA, Hannachi et al. (2007, page 1122, https://doi.org/10.1002/joc.1499), there are many different and ambiguous labels of the components from the literature. We therefore chose to follow the naming from Hannachi et al., 2007, and specified which labels we are using in the introduction of Sect. 3 in the manuscript.

To specify this better in the manuscript, we added to page 4, line 17:

"Many names have been used to describe the output from the EOF analysis (see discussion in Wilks, 2006; Hannachi et al., 2007)."

**Comment 4:** "In my opinion there a too many figures. I'm not saying that they are

needless, but perhaps they can be re-organized or so. In most of them you can find panels with more figures inside. As a result, it's hard to read the axis, the legends, etc."

**Action 4:** We will improve the figures in the revised document by better merging the plots, and making the labels easier to read.
* * *
[Figure]

Atmos. Meas. Tech. Discuss.,
doi:10.5194/amt-2017-226-AC6, 2017

[Figure]

We thank the reviewer for the constructive comments helping to clarify the content of the paper. Please see our responses below.

**Comment 1:** "The paper describes the EOF method in detail, however, does not provide enough justification of using this method and what is the benefits of this method, compared to the conventional anomaly analysis. For example, normalized temperature anomaly time series at each level also can represent the relative strength of the tem-

[Figure]

perature variation at a given level, similar to that shown with EOFs. A few reasons can be included:

1. The EOF analysis can extract the major modes from small scale spacial and temporal variations. This is an important aspect of the EOF method, which can screen out sampling errors from irregular sampled data, i.e., GNSS RO observations.

2. The EOF analysis can be used to explore the structure of the variability within a data set in an objective way, and to analyze relationships within a set of variables. In this study, EOF2 method account the temperature at each level as independent set of variables and connect their modes to analyze the vertical structures of the temperature variation.

"

**Response 1:** We agree that the motivation for using the EOF method can be augmented in the manuscript. Thank you for providing examples for improvement.

Concerning the benefits of EOF analysis compared to the temperature anomalies: Using the temperature anomaly time series directly to represent the temperature variation would require averaging the temperature field, which may cancel out certain patterns. In contrast, the EOF method acts on the whole input field.

See also response to Comment 2 from Referee #1, RC1.

We added to page 4, line 14:

"The EOF analysis can extract major modes from spatial and temporal variations."

**Comment 2:** "The meaning of eigenvectors can be explained to further use them in the reconstruction of the anomaly field."

**Response 2:** We changed page 5, line 7 from

"The output of the EOF analysis is a set of EOFs (EOF$_{out}$), PCs (PC$_{out}$), and their corresponding eigenvalues ($\lambda$)."

to

"The output of the EOF analysis is a set of EOFs (eigenvectors, in this context called EOF$_{out}$), PCs (PC$_{out}$), and their corresponding eigenvalues ($\lambda$)."

**Comment 3:** "The paper shows the linear correlations between PCs and ENSO, QBO, and F10.7 indices, respectively. I recommend to show those conventional indices along with newly suggested PCs from M2 at given levels (e.g., surface, 50hPa) in the your PC plots. It will be an effective way to demonstrate the similarities and differences of new indices."

**Response 3:** We agree that providing plots for the conventional indices will make it easier to compare with the ones suggested in the manuscript.

We added the corresponding time series from Niño 3.4 SST index, the QBO 30 hPa index, and the QBO 50 hPa index in Fig. 3 and Fig. 5.

[revised manuscript text omitted]